# Association between obstructive sleep apnoea and cancer: a cross-sectional, population-based study of the DISCOVERY cohort

Andreas Palm ![ORCID],[1,2] J Theorell-Haglöw,[1] Johan Isakson ![ORCID],[3] Mirjam Ljunggren,[1] Josefin Sundh ![ORCID],[4] Magnus Per Ekström ![ORCID],[5] Ludger Grote[6]

For numbered affiliations see end of article.

**Correspondence to**
Dr Andreas Palm;
andreas.palm@medsci.uu.se

## ABSTRACT

**Objectives** Nocturnal hypoxia in obstructive sleep apnoea (OSA) is a potential risk factor for cancer. We aimed to investigate the association between OSA measures and cancer prevalence in a large national patient cohort.

**Design** Cross-sectional study.

**Settings** 44 sleep centres in Sweden.

**Participants** 62 811 patients from the Swedish registry for positive airway pressure (PAP) treatment in OSA, linked to the national cancer registry and national socioeconomic data (the course of DIsease in patients reported to Swedish CPAP, Oxygen and VEntilator RegistrY cohort).

**Outcome measures** After propensity score matching for relevant confounders (anthropometric data, comorbidities, socioeconomic status, smoking prevalence), sleep apnoea severity, measured as Apnoea-Hypopnoea Index (AHI) or Oxygen Desaturation Index (ODI), were compared between those with and without cancer diagnosis up to 5 years prior to PAP initiation. Subgroup analysis for cancer subtype was performed.

**Results** OSA patients with cancer (n=2093) (29.8% females, age 65.3 (SD 10.1) years, body mass index 30 (IQR 27–34) kg/m$^2$) had higher median AHI (n/hour) (32 (IQR 20–50) vs 30 (IQR 19–45), n/hour, p=0.002) and median ODI (n/hour) (28 (IQR 17–46) vs 26 (IQR 16–41), p<0.001) when compared with matched OSA patients without cancer. In subgroup analysis, ODI was significantly higher in OSA patients with lung cancer (N=57; 38 (21–61) vs 27 (16-43), p=0.012)), prostate cancer (N=617; 28 (17–46) vs 24, (16–39)p=0.005) and malignant melanoma (N=170; 32 (17–46) vs 25 (14–41),p=0.015).

**Conclusions** OSA mediated intermittent hypoxia was independently associated with cancer prevalence in this large, national cohort. Future longitudinal studies are warranted to study the potential protective influence of OSA treatment on cancer incidence.

## INTRODUCTION

Patients with obstructive sleep apnoea (OSA) have recurring airway collapses during sleep with corresponding episodes of hypoxia. Animal models suggest an association between intermittent hypoxia and increased tumour growth, increased angiogenesis, changes in immune function and increases

---

## STRENGTHS AND LIMITATIONS OF THIS STUDY

⇒ The size of the cohort with a large number of patients with verified cancer diagnosis and concomitant obstructive sleep apnoea (OSA).

⇒ High external validity since the Swedevox registry reflects almost 90% of new continuous positive airway pressure treatment starts in Sweden initiated by 44 sleep centres across the country and the cancer registry has almost 100% coverage.

⇒ High internal validity since the Swedevox registry has been validated against medical record data entries and showed more than 95% correctness for age, body mass index and Apnoea-Hypopnoea Index values.

⇒ Important lifestyle risk factors for cancer including smoking, physical activity or food preferences have not been captured on an individual level in our study, but we included data on smoking prevalence in the different regions of Sweden to capture the potential influence of smoking on cancer prevalence.

⇒ Due to the cross-sectional design, our study cannot inform about the potential causality in the association between OSA and cancer.

---

in inflammation and oxidative stress.[1] Several studies investigated the potential causal relationship between OSA and cancer[2–10] and some but not all report an increased prevalence of cancer in the OSA population studied. A recent meta-analysis provided evidence for a 40%–50% increased risk ratio for cancer incidence in OSA patients.[11]

As OSA is associated with several other risk factors for cancer diseases, such as obesity, cardiometabolic disease and lifestyle factors, the independent influence of OSA-related hypoxia for an increased cancer risk remained unsolved. The validity of existing data may be in part hampered by the relatively low number of reported cancer cases, a potential reporting bias due to patient-reported cancer diagnosis without the validation of

independent sources, as well as by a selection bias due to single-centre studies.

In Sweden, 85% of patients with OSA receiving continuous positive airway pressure (CPAP) are reported to the Swedevox registry[12 13] and almost all patients diagnosed with cancer are reported to the mandatory governmental Cancer Registry.[14] The Swedish system with personal identity numbers[15] allows for cross-linkage of various registries, creating large databases with reliable data quality. Thereby, our current study aimed to overcome some of the limitations mentioned above by exploring the association between measures of OSA and cancer prevalence in a large, multicentric national cohort of OSA patients referred for PAP treatment using validated data from the national cancer registry for verified cancer diagnosis and subclassification.

## METHODS

### Study design and population

This was a national, population-based, cross-sectional study from the 'course of DIsease in patients reported to Swedish CPAP, Oxygen and VEntilator RegistrY' cohort. The study has been detailed elsewhere.[13] In this study, data on patients aged ≥16 years with OSA on CPAP therapy, correctly and prospectively reported to the Swedevox registry[12] between 1 July 2010 and 31 December 2017 were cross-linked with data from the mandatory National cancer registry[16] for information about cancer diagnosis, the Prescribed Drug Registry with information about all prescriptions of medications[17] and the National Patient Registry (NPR) covering data and diagnosis from all inpatient and outpatient visits at governmental hospitals.[18] Socioeconomic data (marital status, education level) were covered from Statistics Sweden.[19] An overview of the study flow is presented in figure 1.

### End-point variables and covariables

The primary variable of interest was captured by the diagnose codes 'all-cause cancer' (International classification of Disease version 10 (ICD-10) code C00-99) up to 5 years prior initiation of CPAP therapy. Additional information was captured from organ specific cancer: ear, neck and throat cancer (ENT) (C1–14), colon (C18), lung (C34), malignant melanomas (C43), breast (C50), prostate (C61), urinary (C66–67) and malignant brain tumours (C71). Information on 'skin cancer' (C44) other than malign melanoma was excluded.

Information on sex, body mass index (BMI), Apnoea–Hypopnoea Index (AHI), Oxygen Desaturation Index (ODI), Epworth Sleepiness Scale score (ESS) and date of initiation of treatment were derived from the Swedevox registry.[12 13] Comorbid heart failure (ICD-10 codes I11, I42 and I50) and ischaemic heart disease (ICD-10 code I20–25) was derived from NPR.[18] Comorbid obstructive lung disease (OLD) was classified according to accepted practice by ≥2 collections of antiobstructive drugs (Anatomic Therapeutic Chemical classification system (ATC) code R03) ≤12 months prior to initiation of CPAP and comorbid diabetes was classified by ≥1 collection of antidiabetic drugs (A10) ≤6 months prior initiation of CPAP therapy was registered in the national Prescribed Drug Registry.[17] Information on educational level was obtained from Statistic Sweden and was categorised

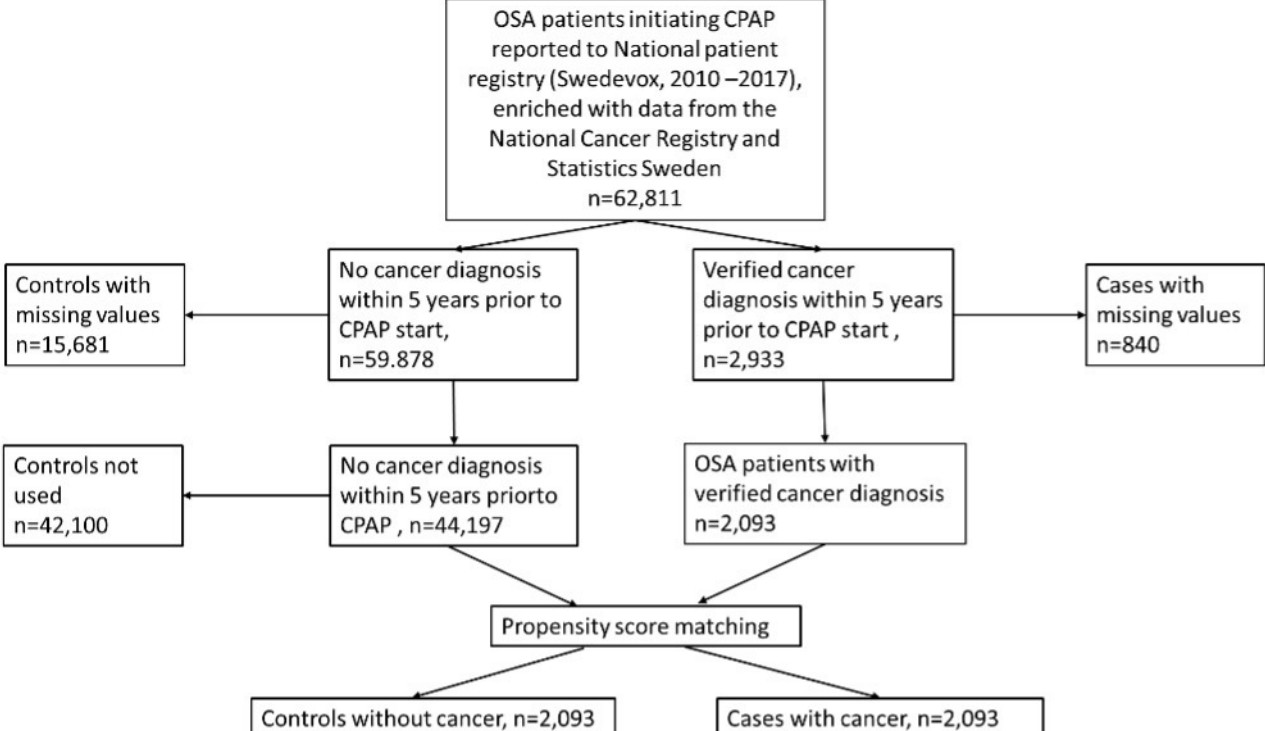

**Figure 1** Study flow chart. CPAP, continuous positive airway pressure; OSA, obstructive sleep apnoea

**Table 1** Characteristics of unmatched and propensity score matched study population

| | | Unmatched | | Matched |
| --- | --- | --- | --- | --- |
| | No cancer | All-cause cancer | No cancer | All-cause cancer |
| | N=59878 | N=2933 | N=2093 | N=2093 |
| Sex, females (%) | 17 647 (29.5%) | 903 (30.8%) | 661 (31.6%) | 623 (29.8%) |
| Age, years | 56.8 (12.5) | 65.5 (10.2) | 65.3 (10.2) | 65.3 (10.1) |
| BMI, kg/m2 | 31 (28–35) | 30 (27–34) | 30 (27–34) | 30 (27–34) |
| ESS, units | 10.3 (5.0) | 9.9 (4.9) | 9.6 (4.9) | 10.0 (4.9) |
| Ischaemic heart disease | 5671 (9.5%) | 425 (14.5%) | 311 (14.9%) | 309 (14.8%) |
| Heart failure | 3263 (5.4%) | 277 (9.4%) | 180 (8.6%) | 186 (8.9%) |
| Diabetes mellitus | 8548 (14.3%) | 526 (17.9%) | 360 (17.2%) | 366 (17.5%) |
| OLD | 7610 (12.7%) | 462 (15.8%) | 325 (15.5%) | 327 (15.6%) |
| Civil status | | | | |
| Married | 31 926 (53.5%) | 1771 (61.0%) | 1300 (62.1%) | 1280 (61.2%) |
| Unmarried | 14 675 (24.6%) | 392 (13.5%) | 289 (13.8%) | 275 (13.1%) |
| Divorced | 10 561 (17.7%) | 520 (17.9%) | 340 (16.2%) | 378 (18.1%) |
| Widower/widow | 2529 (4.2%) | 219 (7.5%) | 164 (7.8%) | 160 (7.6%) |
| Level of education | | | | |
| Low (≤9 years) | 10 547 (21.7%) | 605 (26.1%) | 540 (25.8%) | 546 (26.1%) |
| Medium (10–12 years) | 25 065 (51.5%) | 1061 (45.8%) | 977 (46.7%) | 959 (45.8%) |
| High (≥13 years) | 13 060 (26.8%) | 653 (28.2%) | 576 (27.5%) | 588 (28.1%) |

Matched by anthropometric variables (sex, age, BMI), comorbidities at baseline (ischaemic heart disease, heart failure, diabetes mellitus and obstructive lung disease), socioeconomic factors (civil status and educational level), start year of CPAP treatment and county trichotomised by smoking prevalence. Data are presented as mean (SD) or median (IQR) for continuous measures, and n (%) for categorical measures.
BMI, body mass index; CPAP, continuous positive airway pressure; ESS, Epworth Sleepiness Scale; OLD, obstructive lung disease.

as low (≤9 years), medium (10–12 years) or high (≥13 years), corresponding to compulsory school, secondary school and postsecondary school (college and university), respectively. The Swedevox registry does not contain information about individual smoking data. A marker for smoking exposure was generated from information on annual smoking prevalence between 2006 and 2018, in each county according to Statistics Sweden's Living Conditions Surveys (Undersökningen av levnadsförhållanden/ Survey on Income and Living Conditions, ULF/SILC).[20] Counties were trichotomised according to mean smoking rate (8.3%–12.0%, 12.1%–13.5% and 13.6%–14.8%).

### Statistical analyses

OSA severity was graded by the AHI and ODI both as continuous and categorical variables (AH! <5, 5 to <15 (mild OSA), 15 to <30 (moderate OSA) and ≥30 (severe OSA)). Clinical cut-off values for ODI severity are not defined, and we, therefore, classified ODI into quartiles when used as a categorical variable. Association between sleep recording variables and all-cause cancer diagnosis ≤5 years prior to the initiation of CPAP therapy was analysed. In detail, the relationship between prevalent all-cause cancer and organ-specific cancer and AHI and ODI were analysed using propensity score matched models.[21] For propensity score matching, independent covariables were used as anthropometrics (age, sex, BMI), presence of

comorbidities at baseline (ischaemic heart disease, heart failure, diabetes, OLD), socioeconomic data (civil status and educational level), start year of CPAP treatment and smoking prevalence (by county, trichotomised). These factors were chosen based on direct acyclic graphs using the browser-based environment DAGitty (www. dagitty. net).[22] In addition, we performed a parallel analysis applying logistic regression analysis in the entire baseline population to predict 'all-cause cancer diagnosis' independent of all mentioned covariates used for the propensity matching procedure.

Normally distributed continuous data were expressed as mean±SD, and skewed distributed data were expressed as median with IQR. Categorical data were presented as frequencies and percentages. Differences between groups were analysed using $\chi^2$ test for categorical variables and Student's t-test for continuous variables. A p<0.05 was considered statistically significant. Statistical analyses were conducted using the software packages Stata, V.17.0 (StataCorp).

### Patient and public involvement

None.

**Table 2** OSA severity in propensity score matched patients with OSA stratified by the presence of cancer diagnosis ≤5 years prior to initiation of CPAP treatment

| | No cancer | All-cause cancer | P value |
|---|---|---|---|
| | N=2093 | N=2093 | |
| AHI, events/hour | 30 (19–45) | 32 (20–50) | 0.002 |
| AHI category, events/hour | | | 0.19 |
| AHI <5 | 15 (0.7%) | 14 (0.7%) | |
| AHI 5–14.9 | 272 (13.0%) | 247 (11.8%) | |
| AHI 15–29.9 | 718 (34.3%) | 682 (32.6%) | |
| AHI ≥30 | 1088 (52.0%) | 1150 (54.9%) | |
| ODI, events/hour | 26 (16–41) | 28 (17–46) | <0.001 |
| ODI, quartiles, events/hour | | | 0.002 |
| ODI 0–15.9 | 501 (23.9%) | 433 (20.7%) | |
| ODI 16–27.9 | 587 (28.0%) | 576 (27.5%) | |
| ODI 28–44.9 | 569 (27.2%) | 546 (26.1%) | |
| ODI ≥45 | 436 (20.8%) | 538 (25.7%) | |

Matched by anthropometric variables (sex, age, BMI), comorbidities at baseline (ischaemic heart disease, heart failure, diabetes mellitus and obstructive lung disease), socioeconomic factors (civil status and educational level), start year of CPAP treatment and county trichotomised by smoking prevalence. Data are presented as median (IQR) for continuous measures, and n (%) for categorical measures.
AHI, Apnoea-Hypopnoea Index; BMI, body mass index; CPAP, continuous positive airway pressure; ODI, oxygen desaturation index; OSA, obstructive sleep apnoea.

## RESULTS
### Patient characteristics
Clinical data, in particular age and frequency of comorbidities varied significantly between unmatched cases (N=2933) and controls (N=59 878). The final propensity score matched study population consisted of 2096 patients with cancer (29.8% females, mean age 65.3 (SD 10.1) years, median BMI 30 (IQR 27–34) kg/m$^2$) and 2096 matched OSA patients without cancer (table 1). Totally in the cohort, 91 patients had lung cancer, 46 ENT, 798 prostate cancer, 239 breast cancer, 191 colon cancer, 134 urinary cancer, 254 malignant melanoma and 16 malignant brain tumours ≤5 years prior to CPAP initiation.

After matching procedure, clinical characteristics did not differ between cases and controls except for the ESS score which was slightly elevated in cases compared with controls (ESS score 10.0 (SD 4.9) vs 9.6 (SD 4.9), p=0.02, respectively).

### Sleep apnoea severity in OSA patients with and without cancer
Cases with cancer disease had higher AHI (median 32 (IQR 20–50) vs 30 (19–45), n/hour, p=0.002) and ODI (28 (17–46) vs 26,[16–41] p<0.001) when compared with matched controls without cancer (table 2 and figure 2). Severe nocturnal intermittent hypoxia (ODI quartile IV (>45 n/hour)) was significantly more prevalent in cases (25.7%) compared with controls (20.8%), p=0.001 for ODI categories. Corresponding numbers for the highest AHI class (≥30 n/hour) were 54.9% in cases and 52.4% in controls but the differences in AHI categories did not reach statistical significance.

Subgroup analysis confirmed that ODI was significantly higher in OSA patients with lung cancer (N=57, median 38 (IQR 21–61) vs 27 (16–43), p=0.012)), prostate cancer (N=617, 28 (17–46) vs 24,[16–39] p=0.005) and malignant melanoma (N=170, 32 (17–46) vs 25,[14–41] p=0.015) when compared with corresponding OSA patients without

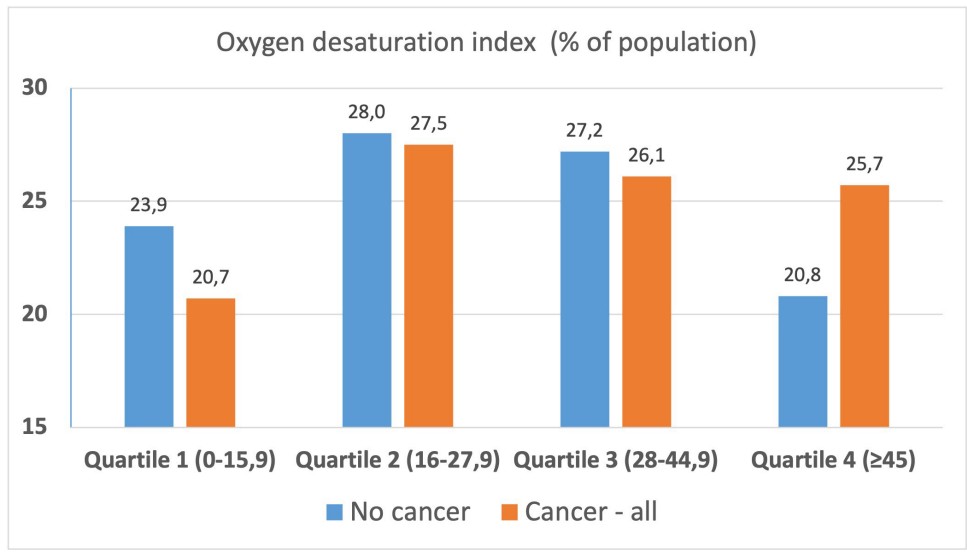

**Figure 2** Distribution of OSA patients with (red bars) and without (blue bars) cancer diagnosis in the four ODI quartiles (x-axis). Propensity score matched analysis in 2093 cases and 2093 controls, ODI distribution differs significantly between cases and controls, p<0.001. ODI, Oxygen Desaturation Index; OSA, obstructive sleep apnoea.

cancer (table 3). The remaining cancer subtypes did not differ in OSA severity measures.

In a parallel analysis including the entire cohort (N=62 811), multivariable logistic regression analysis with cancer as the independent variable, and adjustment for the same variables as in the propensity score matching, we identified significant ORs for ODI and AHI (1.05 (1.03–1.07) and 1.04 (1.02–1.07) per 10 units, p<0.001, respectively) confirming the data obtained by propensity score matching.

## DISCUSSION
### Main findings
Our study identified several important findings. First, OSA is associated with cancer prevalence independent of a substantial number of important confounders. The added risk is small and predominantly seen in patients with severe intermittent hypoxia during sleep measured by ODI rather than in those with frequent respiratory events during sleep measured by AHI. Second, the association between intermittent hypoxia and cancer was identified in lung cancer, malignant melanoma and prostate cancer. Our study underlines the importance of further studies to investigate if preventing and treating sleep apnoea may have beneficial effects on cancer incidence and any outcomes of cancer treatment.

### Cancer prevalence in other cohort studies
Results from previous studies on OSA and cancer are conflicting. One early Spanish multicentre study (4910 patients) identified an increased cancer incidence in OSA patients with an overall OR of 1.45 (95% CI 1.1 to 1.9).[2] In a Canadian cohort study (n=10 149) from a single sleep clinic at an academic hospital, 520 patients had a cancer diagnosis at baseline and 627 patients of the cancer-free patients at baseline were reported with incident cancer at follow-up (7.8 years).[10] No association between the severity of OSA and cancer was found for prevalent or incident cancer. In a twice as large European cross-sectional study (n=19 556 of whom 388 self-reported cancer diagnosis at the time of sleep study), an association between AHI and ODI was found in women but not in men.[8] In a recent French cohort study (n=8748, 5.8-year follow-up, 718 incident cancer cases), an association between nocturnal hypoxia, defined as the per cent of the night spent with saturation <90% and incident cancer was found while there were no associations between AHI or ODI and incident cancer after adjustment for confounders.[9] A recent meta-analysis of 12 studies, including 184 915 patients with OSA (n=75 367 and 3805 cancer cases) and patients without OSA (n=109 548 and 2110 cancer cases) identified a dose-dependent increased risk of incident all-cause cancer with an OR of 1.14 (95% CI 1.04 to 1.25, p=0.006) for mild OSA, 1.36 (95% CI 1.32 to 1.92;p<0.001) for moderate OSA and 1.59 (95% CI 1.45 to 1.74; p<0.001) for severe OSA.[11] Unfortunately, this meta-analysis did not evaluate the association between the amount of OSA-related hypoxic measures and cancer incidence. In this context, our cross-sectional data in one of the largest OSA patient populations so far is in line with previous findings. OSA appears to be associated with cancer, but effect sizes are generally small, and large cohorts are needed to detect any independent effect of OSA on cancer incidence. It cannot be excluded that other confounders, not yet considered in any of the published analyses, may further explain part of the effect sizes accounted for OSA. Notably, several studies show that OSA measures reflecting hypoxic burden appear to better predict the risk for cancer when compared with AHI as traditional OSA severity marker.[8 9] Due to the exact classification of cancer type through the Swedish national registry, we could perform subtype analysis on the influence of OSA-related cancers. We identified more severe OSA in lung cancer, malignant melanoma and in prostate cancer which is in line with previous study results.[23–26] In contrast, we did not identify an increased prevalence of OSA in patients with breast cancer.[27 28]

### Potential mechanisms between the association of OSA and cancer
Previous studies have explored potential mechanistic pathways for the association between OSA and cancer. Experimental studies mimicking OSA by using an intermittent hypoxia model show that intermittent hypoxia promotes incident tumour cell transition, tumour growth and metastasis. Several in vitro cell models identified intermittent hypoxia to promote tumour cell growth in different cell lines.[29] In vivo mouse models showed that melanoma metastasis was increased in animals exposed to intermittent hypoxia mimicking severe OSA.[30] Although these data suggest a possible OSA-related mechanistic pathway for cancer growth and aggressiveness, further studies are needed to explore if and how OSA promote the initial occurrence of cancer. In this context, obesity is an important confounder of the association between OSA and cancer. Obesity in the general population has nearly doubled in the last 20 years and by 2017, 17% of the Swedish adult population are obese, defined as BMI exceeding 30 kg/m$^2$.[31] Presence of OSA is closely linked to obesity,[32] and in Sweden, the mean BMI when initiating CPAP therapy is 32 kg/m$^2$.[12] Several cancer forms, that is, breast, colon, kidney, pancreas and oesophageal cancer, are also associated with higher BMI.[33] As both age-standardised cancer incidence[34] and the prevalence of obesity are increasing[31] in parallel, we carefully controlled for both factors in our analysis.

Inflammation may be considered as a third potential pathway for the link between OSA and cancer. OSA increases systemic inflammation with increased levels of high-sensitivity C-reactive protein (hsCRP), Interleukin-6 (IL-6), and Tumour necrosis factor α (TNF-α).[35 36] Systemic inflammation has been linked to the risk of cancer/cancer development but also tumour growth.[37] In addition, a low degree of inflammation has been identified in adipocytes which may contribute to the

**Table 3** Frequency of organ-specific cancer prevalence in patients with OSA ≤5 years prior to the initiation of CPAP in relation to AHI and ODI in propensity score matched cohorts

| | No lung cancer N=57 | Lung cancer N=57 | P value | No ENT cancer N=31 | ENT cancer N=31 | P value | No prostate cancer N=617 | Prostate cancer N=617 | P value |
|---|---|---|---|---|---|---|---|---|---|
| AHI, events/hour | 28 (18–44) | 38 (25–56) | 0.019 | 29 (18–46) | 30 (17–41) | 0.30 | 28 (19–44) | 33 (20–50) | 0.006 |
| AHI category, events/hour | | | 0.10 | | | 0.036 | | | 0.035 |
| AHI<5 | 0 | 0 | | 0 | 0 | | 4 (0.6%) | 2 (0.3%) | |
| AHI 5–14.9 | 8 (14.0%) | 6 (10.5%) | | 1 (3.2%) | 7 (22.6%) | | 70 (11.3%) | 66 (10.7%) | |
| AHI 15–29.9 | 21 (36.8%) | 12 (21.1%) | | 15 (48.4%) | 8 (25.8%) | | 251 (40.7%) | 208 (33.7%) | |
| AHI>30 | 28 (49.1%) | 39 (68.4%) | | 15 (48.4%) | 16 (51.6%) | | 292 (47.3%) | 341 (55.3%) | |
| ODI, events/hour | 27 (16–43) | 38 (21–61) | 0.012 | 24 (15–46) | 23 (13–39) | 0.22 | 24 (16–39) | 28 (17–46) | 0.005 |
| ODI, quartiles, events/hour | | | 0.034 | | | 0.21 | | | 0.009 |
| ODI 0–15.9 | 13 (22.8%) | 6 (10.5%) | | 8 (25.8%) | 11 (35.5%) | | 147 (23.8%) | 129 (20.9%) | |
| ODI 16–27.9 | 17 (29.8%) | 17 (29.8%) | | 10 (32.3%) | 6 (19.4%) | | 210 (34.0%) | 170 (27.6%) | |
| ODI 28–44.9 | 15 (26.3%) | 9 (15.8%) | | 5 (16.1%) | 10 (32.3%) | | 139 (22.5%) | 161 (26.1%) | |
| ODI >45 | 12 (21.1%) | 25 (43.9%) | | 8 (25.8%) | 4 (12.9%) | | 121 (19.6%) | 157 (25.4%) | |

| | No Breast cancer N=162 | Breast cancer N=162 | P value | No colon cancer N=123 | Colon cancer N=123 | P value | No urinary cancer N=93 | Urinary cancer N=93 | P value |
|---|---|---|---|---|---|---|---|---|---|
| AHI, events/hour | 28 (17–46) | 28 (18–46) | 0.49 | 33 (21–53) | 30 (21–48) | 0.41 | 31 (19–47) | 35 (24–52) | 0.15 |
| AHI category, events/hour | | | 0.59 | | | 0.31 | | | 0.40 |
| AHI <5 | 4 (2.5%) | 2 (1.2%) | | 4 (3.3%) | 2 (1.6%) | | 1 (1.1%) | 1 (1.1%) | |
| AHI 5–14.9 | 29 (17.9%) | 24 (14.8%) | | 10 (8.1%) | 13 (10.6%) | | 13 (14.0%) | 6 (6.5%) | |
| AHI 15–29.9 | 51 (31.5%) | 60 (37.0%) | | 35 (28.5%) | 46 (37.4%) | | 26 (28.0%) | 30 (32.3%) | |
| AHI >30 | 78 (48.1%) | 76 (46.9%) | | 74 (60.2%) | 62 (50.4%) | | 53 (57.0%) | 56 (60.2%) | |
| ODI, events/hour | 24 (14–39) | 27 (17–49) | 0.26 | 28 (18–50) | 26 (17–43) | 0.47 | 28 (16–47) | 32 (21–50) | 0.11 |
| ODI, quartiles, events/hour | | | 0.30 | | | 0.71 | | | 0.67 |
| ODI 0–15.9 | 49 (30.2%) | 38 (23.5%) | | 24 (19.5%) | 26 (21.1%) | | 20 (21.5%) | 14 (15.1%) | |
| ODI 16–27.9 | 39 (24.1%) | 49 (30.2%) | | 34 (27.6%) | 36 (29.3%) | | 26 (28.0%) | 25 (26.9%) | |
| ODI 28–44.9 | 40 (24.7%) | 34 (21.0%) | | 28 (22.8%) | 32 (26.0%) | | 22 (23.7%) | 25 (26.9%) | |
| ODI >45 | 34 (21.0%) | 41 (25.3%) | | 37 (30.1%) | 29 (23.6%) | | 25 (26.9%) | 29 (31.2%) | |

| | No malignant melanoma | Malignant melanoma | P value | No malignat brain tumour | Malignant brain tumour | P value |
|---|---|---|---|---|---|---|

**Table 3** Continued

| | No lung cancer | Lung cancer | P value | No ENT cancer | ENT cancer | P value | No prostate cancer | Prostate cancer | P value |
|---|---|---|---|---|---|---|---|---|---|
| | N=57 | N=57 | | N=31 | N=31 | | N=617 | N=617 | |
| | N=170 | N=170 | | N=8 | N=8 | | | | |
| AHI, events/hour | 28 (17–45) | 35 (23–52) | 0.003 | 31 (12–40) | 40 (16–54) | 0.25 | | | |
| AHI category, events/hour | | | 0.072 | | | 0.86 | | | |
| AHI <5 | 2 (1.2%) | 1 (0.6%) | | 0 | 0 | | | | |
| AHI 5–14.9 | 25 (14.7%) | 15 (8.8%) | | 2 (25.0%) | 2 (25.0%) | | | | |
| AHI 15–29.9 | 65 (38.2%) | 53 (31.2%) | | 1 (12.5%) | 1 (12.5%) | | | | |
| AHI >30 | 78 (45.9%) | 101 (59.4%) | | 5 (62.5%) | 5 (62.5%) | | | | |
| ODI, events/hour | 25 (14–41) | 32 (17–46) | 0.015 | 29 (13–34) | 36 (15–56) | 0.19 | | | |
| ODI, quartiles, events/hour | | | 0.13 | | | 0.28 | | | |
| ODI 0–15.9 | 48 (28.2%) | 32 (18.8%) | | 3 (37.5%) | 2 (25.0%) | | | | |
| ODI 16–27.9 | 47 (27.6%) | 45 (26.5%) | | 1 (12.5%) | 1 (12.5%) | | | | |
| ODI 28–44.9 | 44 (25.9%) | 50 (29.4%) | | 4 (50.0%) | 2 (25.0%) | | | | |
| ODI >45 | 31 (18.2%) | 43 (25.3%) | | 0 (0.0%) | 3 (37.5%) | | | | |

Matched by anthropometric variables (sex, age, BMI), comorbidities at baseline (ischaemic heart disease, heart failure, diabetes mellitus and obstructive lung disease), socioeconomic factors (civil status and educational level), start year of CPAP treatment and county trichotomised by smoking prevalence. AHI and ODI are expressed as both continuous and categorised variables.

Data are presented as median (IQR) for continuous measures, and n (%) for categorical measures.

AHI, Apnoea-Hypopnoea Index; BMI, body mass index; CPAP, continuous positive airway pressure; ENT, ear, nose and throat; ODI, Oxygen Desaturation Index; OSA, obstructive sleep apnoea.

overall increased inflammation in obesity. Further studies linking markers of inflammation to cancer incidence in OSA populations are of particular interest.

Our study has several strengths. First, the study has a high validity as we included a large number of patients with a verified cancer diagnosis and concomitant OSA. The National cancer registry has the highest quality due to regularly performed validation studies by the National Social Welfare Board. The sleep apnoea-related data within the Swedevox registry has recently been validated against the medical record data entries and showed more than 95% correctness for age, BMI and AHI values.[38] Second, our study cohort is representative for studying the association between OSA and cancer prevalence. The Swedevox registry reflects almost 90% of new CPAP treatment starts in Sweden initiated by 44 sleep centres across the country.[12] The national cancer registry has nearly 100% coverage.[16] Socioeconomic data are based on information from the National Tax Agency. No patient was lost due to missing data in the national registries. Third, we used a strong analysis methodology with sufficient power due to a large number of cases. The propensity score matching of cases and controls adjusting for many important confounders can be viewed as close to a randomised study setting. A parallel multivariable Cox-regression analysis confirmed the results of the propensity score matched analysis.

Nonetheless study limitations need also to be addressed. First, sleep apnoea within the Swedevox registry is almost exclusively classified by polygraphy and not by polysomnography. However, device methodology is very homogeneous in Sweden and national guidelines for the analysis of polygraphic recordings are published.[39] Polygraphic recording may underestimate the actual AHI classification but to a lesser extent, the ODI count. This may, in part explain why ODI is a stronger predictor of cancer prevalence than AHI. Second, we do not have other measures of nocturnal hypoxia other than ODI, for example, the time spent in saturation below 90% or hypoxic burden analysis of the saturation curve, recently described as a significant predictor of cancer in OSA patients.[2 8 9 40] Third, important lifestyle risk factors for cancer, including smoking, physical activity or food preferences, have not been captured on an individual level in our study. However, we have a strong classification of socioeconomic factors. We also included data on smoking prevalence in the different regions of Sweden to capture the potential influence of smoking on cancer prevalence. Research into the impact of socioeconomic factors on cancer incidence typically find a larger prevalence of solid tumours in groups with less fortunate social standing with breast cancer being the sole outlier. The single most important variable in cancer incidence is tobacco smoking where large differences are seen between socioeconomic groups.[41] In case of lacking information on individual patient data for such risk factors, the use of socioeconomic stratification represents the best available method to adjust for this. Fourth, our study included only PAP-treated OSA patients reflecting

the upper range of OSA severity and not the entire OSA spectrum. This patient selection may have limited our possibility to fully assess the dose response relationship between OSA severity and cancer prevalence. However, the OSA patient group with severe intermittent hypoxia demonstrated the strongest risk increase for cancer prevalence. Finally, due to the cross-sectional study design, our study cannot speculate about the causality in the association between OSA and cancer.

### Clinical implications and research agenda
Our study provides further cross-sectional evidence to the important clinical question of the association between OSA and cancer in OSA patients. Even if we did not analyse longitudinal data on the incidence of cancer, we see a significant yet limited effect size for OSA on the risk for prevalent cancer. More importantly, future analysis needs to focus on the effect of OSA treatment with PAP on cancer incidence and survival.

## CONCLUSION
OSA mediated intermittent hypoxia is an independent risk factor for cancer in this large, national OSA patient cohort. Future longitudinal studies are warranted to study the potential influence of OSA treatment on cancer incidence in patients with OSA.

**Author affiliations**
[1]Department of Medical Sciences, Lung, Allergy and Sleep Research, Uppsala University, Uppsala, Sweden
[2]Centre for Research and Development, Region of Gävleborg Gävle Hospital, Gävle, Sweden
[3]Centre for Research and Development, Region of Gävleborg, Gävle Hospital, Gävle, Sweden
[4]Department of Respiratory Medicine, Faculty of Medicine and Health, Örebro University, Örebro, Sweden
[5]Department of Clinical Sciences, Respiratory Medicine and Allergology, Lund University, Lund, Sweden, Lund, Sweden
[6]Sahlgrenska Academy, Gothenburg University, Centre for Sleep and Wake Disorders, Goteborg, Sweden

**Contributors** AP, JT-H, JI, ML, JS, MPE and LG contributed to the conception and design of the study. AP performed statistical analyses, and all authors verified the underlying data. AP and LG wrote the first draft. All authors had full access to all the data in the study and had final responsibility for the decision to submit for publication. All authors participated in data interpretation, drafting of the manuscript and final approval for submission. AP is responisble for the overall content as a guarantor.

**Funding** AP was supported by grants from Gävle Cancer Foundation (grant number N/A), the Swedish Society for Sleep Research and Sleep Medicine (grant number N/A), Centre for Research and Development, Uppsala University/Region Gävleborg (grant number N/A), Bror Hjerpstedt's Foundation (grant number N/A) and Uppsala Heart and Lung Foundation (grant number N/A) and Regional Research Council in Mid Sweden (RFR-931234). JT-H was supported by a grant from Swedish Heart and Lung Foundation (20190607, 20190611). MPE was supported by unrestricted grants from the Swedish Research Council (Dnr 2019-02081). LG was supported by the Swedish Heart and Lung Foundation (20180567, 20210529).

**Competing interests** None declared.

**Patient and public involvement** Patients and/or the public were not involved in the design, or conduct, or reporting, or dissemination plans of this research.

**Patient consent for publication** Not applicable.

**Provenance and peer review** Not commissioned; externally peer reviewed.

**Data availability statement**  Data are available on reasonable request. The steering committee of the Swedevox quality registry will consider reasonable requests for the sharing of deidentified patient-level data. Requests should be made to the corresponding author.

**ORCID iDs**
Andreas Palm http://orcid.org/0000-0002-0590-0417
Johan Isakson http://orcid.org/0000-0001-5787-0072
Josefin Sundh http://orcid.org/0000-0003-1926-8464
Magnus Per Ekström http://orcid.org/0000-0002-7227-5113

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
