## [Reviewer comments · BMJ Open]

ARTICLE DETAILS

TITLE (PROVISIONAL)	Association between obstructive sleep apnoea and cancer: a cross-sectional, population-based study of the DISCOVERY cohort
AUTHORS	Palm, Andreas; Theorell-Haglöw, J; Isakson, Johan; Ljunggren, Mirjam; Sundh, Josefin; Ekström, Magnus; Grote, Ludger

VERSION 1 – REVIEW

REVIEWER	Luis Seijo Clínica Universidad de Navarra
REVIEW RETURNED	11-Jul-2022

GENERAL COMMENTS	A very interesting article confirming the potential role of OSAS in cancer. The article is well written and the design seems appropriate. The study conclusions match existing evidence and the data, including the finding that OSAS is probably associated with some cancers and that the association is relatively modest and requires large samples to prove. A major strength of the findings is the excellent quality of the data obtained from the Swedish registries. I would recommend eliminating from the discussion the comments related to a potential causal role of hypoxia and OSAS in cancer as the authors admit that the data do not support a cause, but simply state an association, and furthermore, this was not the purpose of their study. I would focus more on discussing the association, mentioning recent studies for example focusing on the association between lung cancer and OSAS. They mention the association between T90 and cancer, and the fact that their data do not include this important variable which has appeared over and over as a key variable in recent publications. They acknowledge, for example, that the AHI seems less robust as a marker of risk than hypoxemia. I would ask that they elaborate more on this important distinction. I also think that they should focus more on some important study limitations. For example, their OSAS patients are treated with CPAP, so the study excludes untreated individuals with OSAS, and the true incidence of smoking, a key variable in lung cancer, is inferred from national prevalence data rather than the registries themselves. At least for lung cancer patients these data are probably known.
---

REVIEWER	Frederic Roche PRES Univ Lyon
REVIEW RETURNED	02-Aug-2022

GENERAL COMMENTS	The present study focused on an interesting and topical pathophysiological and epidemiological question: is obstructive sleep apnea syndrome associated with an increase in the
---

	prevalence of cancers and, if so, which cancers would be more related to the stress of this condition? This is a good quality, multicenter Swedish epidemiological study that crossed prevalence registers (cancer and apnea). The authors then carried out case-control studies for different types of cancer. The results speak for themselves and invite further reflection which, despite preclinical studies, has been somewhat sidelined due to negative statistical linkage results in various European and North American cohorts. The limitations of the study are well addressed and the authors are well aware of the relative significance of their results, given the purely cross-sectional nature of the study, the lack of longitudinal follow-up, and the fact that risk factors common to both diseases, such as smoking, are not taken into account. Nevertheless, the work is well written. The pathophysiological part of the discussion would deserve more scientific substance (the data in the literature are abundant). There remain for me some additional points to clarify in a future version of this paper The first is the lack of more data on hypoxemic load: the ODI is not sufficient to reliably represent this load. Authors should have access to the time spent below 90% (SpO2) and the minimum and mean values of this SpO2 (data available in all ventilatory polygraph systems. It is not known whether patients with cancer already diagnosed at the time of CPAP were included in the analysis. If so, what are the results of this sub-study? We understand that the period of interest for the search for cancer was 5 years after the installation of CPAP: why was this time frame chosen? The authors talk about longitudinal data within the limits of the study: do they have a longer follow-up? These are patients who all received CPAP for their OSA: were the patients who received a mandibular advancement orthosis not included? The selection bias appears to be important. The authors (I may have missed the information) did not focus on primary brain tumors: why: there are interesting Korean epidemiological data on this subject. Finally, tables 1 and 2 are not in the right reading direction and should be rotated (horizontalized). I am not a statistician but the selection of the "control" subjects for each of the cancer typologies did not appear very clear to me. All in all, this is a good paper and very interesting in the results it brings to the body of literature in the field. We will wait impatiently for the longitudinal data of these registers!
--	---

VERSION 1 – AUTHOR RESPONSE

Reviewer: 1

Dr. Luis Seijo, Clínica Universidad de Navarra

Comments to the Author:

A very interesting article confirming the potential role of OSAS in cancer. The article is well written and the design seems appropriate. The study conclusions match existing evidence and the data, including the finding that OSAS is probably associated with some cancers and that the association is relatively modest and requires large samples to prove. A major strength of the findings is the excellent quality of the data obtained from the Swedish registries.

We thank the reviewer for the positive feedback.

Comment 1.1

I would recommend eliminating from the discussion the comments related to a potential causal role of hypoxia and OSAS in cancer as the authors admit that the data do not support a cause, but simply state an association, and furthermore, this was not the purpose of their study. I would focus more on discussing the association, mentioning recent studies for example focusing on the association between lung cancer and OSAS. They mention the association between T90 and cancer, and the fact that their data do not include this important variable which has appeared over and over as a key variable in recent publications. They acknowledge, for example, that the AHI seems less robust as a marker of risk than hypoxemia. I would ask that they elaborate more on this important distinction.

We agree with the reviewer that this paper aims to analyze the association between cancer, various cancer types, and OSA, not to explore any potential mechanisms in detail. One abundant sentence and one reference from the discussion section “Hypoxia inducible factor (HIF) may activate vascular endothelial growth factor (VEGF) and promote vascular angiogenesis important for tumor growth (29)” have been deleted. We also fully agree with the reviewer that the absence of the variable T90 is a major limitation of our study as more and more data emphasize the significance of this variable in capturing the degree of hypoxic burden in OSA. A reference to a recently published meta-analysis (Tan et al) has been added to the discussion. Unfortunately, as mentioned in the discussion section, the T90 variable is not reported to the Swedevox registry. This is stated in the limitations section of the discussion, but by adding the phrase “other than ODI” to the sentence “Second, we do not have other measures of nocturnal hypoxia, other than ODI, for example, the time spent in saturation below 90% or hypoxic burden analysis of the saturation curve”, we hope that this limitation of our study now is satisfactorily emphasized. We have plans to add this variable to the registry next time we will revise the variable list in the Sleep Apnea Registry.

Comment 1.2

I also think that they should focus more on some important study limitations. For example, their OSAS patients are treated with CPAP, so the study excludes untreated individuals with OSAS, and the true incidence of smoking, a key variable in lung cancer, is inferred from national prevalence data rather than the registries themselves. At least for lung cancer patients, these data are probably known.

We again thank the reviewer for identifying important limitations of our study. The study group in this study is patients with moderate to severe OSA starting CPAP therapy. The time window for the current analysis is up to five years prior to the start of CPAP therapy. Our data reflect cancer risk in untreated OSA before the start of CPAP treatment. In an ongoing extension of our DISCOVERY database, we will also add the Swedish diagnosis-based Swedish Sleep Apnea Registry (SESAR) including the entire spectrum of OSA severity from very mild to severe OSA. In this extended cohort, we plan to perform a longitudinal study and analyze incident cancer and the associations with OSA severity measures as well as the impact of different treatment modalities/CPAP adherence rates on cancer risk.

Unfortunately, information about smoking status is not reported to the Swedevox registry and was not available to us by any other registry in the DISCOVERY-cohort either. In the analysis, we partially compensate for that limitation by adding socio-economic status variables and information about smoking prevalence by the county to our analysis.

In order to further explain our study limitations we extended the corresponding chapter in the discussion of the revised manuscript.

Reviewer: 2

Dr. Frederic Roche, PRES Univ Lyon

Comments to the Author:

The present study focused on an interesting and topical pathophysiological and epidemiological question: is obstructive sleep apnea syndrome associated with an increase in the prevalence of cancers and, if so, which cancers would be more related to the stress of this condition?

This is a good quality, multicenter Swedish epidemiological study that crossed prevalence registers (cancer and apnea). The authors then carried out case-control studies for different types of cancer. The results speak for themselves and invite further reflection which, despite preclinical studies, has been somewhat sidelined due to negative statistical linkage results in various European and North American cohorts.

The limitations of the study are well addressed, and the authors are well aware of the relative significance of their results, given the purely cross-sectional nature of the study, the lack of longitudinal follow-up, and the fact that risk factors common to both diseases, such as smoking, are not taken into account.

Thank you for the kind comments on our manuscript.

Nevertheless, the work is well written. The pathophysiological part of the discussion would deserve more scientific substance (the data in the literature are abundant). There remain for me some additional points to clarify in a future version of this paper

Response: We agree even with reviewer one suggesting to concentrate on our study type as a cross-sectional analysis of a large patient cohort and not to speculate on the potential mechanisms we could not address in our data format of a national patient registry.

Comment 2.1

The first is the lack of more data on hypoxemic load: the ODI is not sufficient to reliably represent this load. Authors should have access to the time spent below 90% (SpO₂) and the minimum and mean values of this SpO₂ (data available in all ventilatory polygraph systems).

We thank the reviewer for this important remark which was also pointed out by Reviewer 1. We agree with the reviewer that the lack of the variable T90 is a major limitation of our study. Unfortunately, the information on other hypoxia variables than ODI is currently not in our available registry data. We have now expanded our discussion of study limitations as follows: "Second, we do not have other measures of nocturnal hypoxia other than ODI, for example, the time spent in saturation below 90% or hypoxic burden analysis of the saturation curve", As mentioned to reviewer 1, we aim to add variables in our database better reflecting hypoxic burden in the future.

Comment 2.2

It is not known whether patients with cancer already diagnosed at the time of CPAP were included in the analysis. If so, what are the results of this sub-study? We understand that the period of interest for the search for cancer was 5 years after the installation of CPAP: why was this time frame chosen.

Thank you for this question. In this cross-sectional study, we analyzed the relationship between all cancer diagnoses except skin cancer during the time span of five years PRIOR TO, NOT AFTER, initiation of CPAP therapy, and OSA severity measures. We further clarified the study design in the revised version of the manuscript. The time window of five years is an arbitrary cut off point and was adapted from other published studies with the same topic. We agree, other time frames may be

eligible as the risk of cancer may evolve over decades. We added this discussion in the section about study limitations.

Comment 2.3

The authors talk about longitudinal data within the limits of the study: do they have a longer follow-up? These are patients whom all received CPAP for their OSA: were the patients who received a mandibular advancement orthosis not included? The selection bias appears to be important.

Thank you for the chance to clarify this. Reviewer 1 also had questions about treatments other than CPAP and its association with cancer. In the present DISCOVERY-cohort, there is a mean follow-up time of 3.0 ± 2.1 years, and the study patients are reported to the Swedish treatment-based Swedevox registry, which means that all OSA patients in the study are those who have started treatment with CPAP. In an ongoing extension of the DISCOVERY-cohort, we will also include patients reported to the diagnosis-based Swedish Sleep Apnea Registry (SESAR) with data until 2022. This means that we will then have the ability to analyse the whole spectrum of patients with OSA, and the mean follow-up time will be longer. We are planning a follow-up on this study with a longitudinal one very soon.

Comment 2.4

The authors (I may have missed the information) did not focus on primary brain tumors: why: there are interesting Korean epidemiological data on this subject.

We have data on the brain tumors, and we initially opted to exclude them due to the rather small number of events. However, as suggested we have now added data about malignant brain tumors to table 3. In addition, we have now added a sentence with information about the number of organ-specific cancers in the cohort to the patient characteristics paragraph.

Comment 2.5

Finally, tables 1 and 2 are not in the right reading direction and should be rotated (horizontalized).

We appreciate the attempt to make our tables 1 and 2 more readable.. We also refer to the the statistical reviewer (see below) who recommended to maintain the overall design of table 1 and 2 but we worked on the final design to make them more readable.

Comment 2.6

I am not a statistician but the selection of the "control" subjects for each of the cancer typologies did not appear very clear to me.

As mentioned in the statistical analysis section, the "control" subjects are propensity score-matched OSA patients without all-cause and organ-specific cancer. We changed the label in the revised version of the manuscript to increase clarity to the reader.

Comment 2.7

All in all, this is a good paper and very interesting in the results it brings to the body of literature in the field. We will wait impatiently for the longitudinal data of these registers!

Thank you. As I mentioned earlier, longitudinal data are in sight!

VERSION 2 – REVIEW

REVIEWER	Frederic Roche PRES Univ Lyon
REVIEW RETURNED	08-Jan-2023

GENERAL COMMENTS

The paper has been seriously and thoroughly reworked. The new version proposed is of good quality and seems to me acceptable for publication. This type of cohort study is indeed essential at the present time to try to demonstrate a link between chronic intermittent hypoxic stress and cancers. This exhaustive data base on a very large number of variables and which is practically on a country scale is a wealth of scientific information even if the causality of the described pathologies is impossible to clarify formally. Similarly, the effect of sleep apnea treatment on certain cancers must be precisely evaluated in the future. Such an article would deserve an associated editorial.